# Liver function indicators in patients with breast cancer before and after detection of hepatic metastases-a retrospective study

Carmen Leser[1]*, Georg Dorffner[2], Maximilian Marhold[3], Anemone Rutter[1], Mert Döger[1], Christian Singer[1], Deirdre Maria König-Castillo[4], Christine Deutschmann[1], Iris Holzer[1], Daniel König-Castillo[5], Daphne Gschwantler-Kaulich[1]

1 Department of Obstetrics and Gynecology, Cancer Comprehensive Center, Medical University of Vienna, Vienna, Austria, 2 Section for Artificial Intelligence and Decision Support, Medical University of Vienna, Vienna, Austria, 3 Medical University of Vienna Department of Medicine I, Clinical Division of Oncology, Comprehensive Cancer Center, Medical University of Vienna, Vienna, Austria, 4 Department of Urology, Comprehensive Cancer Center, Medical University of Vienna, Vienna, Austria, 5 Clinical Division of Social Psychiatry, Department of Psychiatry and Psychotherapy, Medical University of Vienna, Vienna, Austria

* carmen.leser@meduniwien.ac.at

**Data Availability Statement:** All relevant data are within the paper and its Supporting Information files.

## Abstract

### Background

Liver metastases are common in patients with breast cancer, and determining the factors associated with such metastases may improve both their early detection and treatment. Given that liver function protein level changes in these patients have not been determined, the aim of our study was to investigate liver function protein level changes over time, spanning 6 months before the detection of liver metastasis to 12 months after.

### Methods

We retrospectively studied 104 patients with hepatic metastasis from breast cancer who were treated at the Departments of Internal Medicine I and the Department of Obstetrics and Gynecology at the Medical University of Vienna between 1980 and 2019. Data were extracted from patient records.

### Results

Aspartate aminotransferase, alanine aminotransferase, gamma-glutamyltransferase, lactate dehydrogenase and alkaline phosphatase were significantly elevated when compared to normal range 6 months before the detection of liver metastases ($p<0.001$) Albumin was decreased ($p<0.001$). The values of aspartate aminotransferase, gamma-glutamyltransferase, and lactate dehydrogenase were significantly increased at the time of diagnosis compared to 6 months prior ($p<0.001$). Patient- and tumor-specific parameters had no influence on these liver function indicators. Elevated aspartate aminotransferase ($p = 0.002$) and reduced albumin ($p = 0.002$) levels at the time of diagnosis were associated with shorter overall survival.

**Funding:** The author(s) received no specific funding for this work.

**Competing interests:** The authors have declared that no competing interests exist.

## Conclusion

Liver function protein levels should be considered as potential indicators when screening for liver metastasis in patients with breast cancer. With the new treatment options available, it could lead to prolonged life.

## Introduction

Breast cancer accounts for 30% of all malignancies in women with a mortality rate second only to that of lung cancer in the United States [1] and causes the highest number of cancer-related deaths among women globally [2]. While the prognosis is generally favorable owing to the advances in medical treatments achieved over recent decades, 20–30% of patients with breast cancer develop secondary metastases [3–5]. The incidence of metastases has been rising, and once it occurs, the patient's prognosis deteriorates significantly; the 5-year survival rate drops from 80% to 23% [6–8].

The impact of metastases on survival signifies the importance of their early detection, as prompt intervention can improve patient outcomes. The liver is one of the most common sites of metastasis in patients with breast cancer; furthermore, the 5-year survival rate of patients who develop hepatic metastases is only 8.5%, which is one of the poorest [9, 10]. When performed early, surgical treatment of liver metastases has been shown to be effective in prolonging life expectancy [11, 12]; as such, the early detection of liver metastases is of high importance.

Previous studies have found that liver function is poor in 92% of all patients diagnosed with breast cancer liver metastasis (BCLM), with gamma-glutamyl transferase (GGT) and alkaline phosphatase (AP) showing the strongest positive correlations [13]. Cao et al. showed that stage III cancer or c-erbB-2-positivity are positively associated with liver metastases [14]. Moreover, alanine aminotransferase (ALT), aspartate aminotransferase (AST), GGT, AP, lactate dehydrogenase (LDH), and cancer antigen 15–3 levels are significantly higher in patients with BCLM than in those without [10, 11]. Thus, tumor marker and liver function tests combined may be helpful in screening parameters for BCLM. Furthermore, serum albumin and total bilirubin levels can help predict the survival of patients with liver metastases [14, 15].

The purpose of this study was to identify serum levels of liver function-associated proteins (via common screening tests) that would predict BCLM even before symptoms occur or metastatic lesion are detected. To the best of our knowledge, our study is the first to investigate liver protein levels not only at the time of BCLM diagnosis and the following months, but also prior to diagnosis. We aimed to investigate serum levels of these proteins during the timeframe of 6 months before diagnosis until 12 months after.

## Methods

### Patient selection

In this retrospective analysis of prospectively collected data, patients between the ages of 18 and 100 years who were treated for breast cancer at the Department of Gynecology and the Department of Internal Medicine I (Oncology) of the Medical University of Vienna between January 1980 and May 2019 were included.

Liver function protein levels measured 6 months prior to the diagnosis of BCLM, at the time of diagnosis, and 12 months after diagnosis were acquired. These included AST, ALT,

**Table 1. Normal ranges of liver function proteins.**

|  | Normal range |
|---|---|
| **AST U/L** | <35 |
| **ALT U/L** | <35 |
| **GGT U/L** | <40 |
| **LDH U/L** | <250 |
| **AP U/L** | <105 |
| **Albumin g/L** | 35–52 |
| **Bilirubin mg/dL** | <1.2 |

Normal ranges of liver function proteins as used in the University of Vienna.

AST, aspartate aminotransferase; ALT, alanine aminotransferase; GGT, gamma-glutamyl transferase; LDH, lactate dehydrogenase, AP, alkaline phosphatase.

GGT, LDH, AP, albumin, and bilirubin. Moreover, data on tumors including molecular sub-type (luminal A, luminal B, *HER2*-positive, or triple negative), tumor type (invasive ductal carcinoma or invasive lobular carcinoma), and grading (I–III); age (years); body mass index (BMI in kg/m$^2$); history of smoking (yes/no); history of alcohol consumption (yes/no); and whether any prescribed medications were potentially toxic to the liver (yes/no) were collected. Normal ranges of values were defined as our institution's local standard (Table 1). The study was approved by a local ethics committee (number EK 1488/2019).

## Statistical analysis

To test elevated levels of liver function as measured 6 months prior to the diagnosis of BCLM, a chi-square test was used to compare the percentage of cases above the respective normal value with the expected 5% for one-sided ranges (or 2.5% for two-sided ranges, respectively) assuming that upper normality levels mark the 95th (or 97.5) percentile of a distribution of values considered to be normal. Values at the time of diagnosis and 12 months thereafter were compared to values 6 months prior to diagnosis using a Mann-Whitney U-test (owing to the skewed non-normal distribution of each variable). A significance level of 0.05 was corrected by the Bonferroni method given the number of 21 tests, yielding a significance level of $\alpha = 0.0024$.

On a purely exploratory basis, dependencies of liver function protein levels on other variables were tested using either a correlation analysis (Pearson correlation for metrical variables, Spearman correlation for ordinal variables) or a Mann-Whitney U-test (for nominal variables), using a significance level of $\alpha = 0.05$.

## Results

### Patient characteristics

Four hundred patients with metastases arising from breast cancer were screened; after 296 were excluded owing to the lack of liver metastases, 104 patients with BCLM were included in this study. Among them, 69 (66.3%) had serum liver protein data both 6 months before the diagnoses of BCLM and at the time of diagnosis. Furthermore, 74 patients (71.2%) had laboratory data acquired at the time of BCLM diagnosis and 12 months after, and 40 patients (38.5%) had laboratory results available for all 3 time points. The median BMI was 24.4 kg/m$^2$ (18.0–42.9 kg/m$^2$). Five patients were excluded due to a lack of data.

In terms of risk factors for elevated liver function proteins (Fig 1), 98 patients (94.2%) were taking at least one medication with hepatotoxic potential (on average, each patient was taking

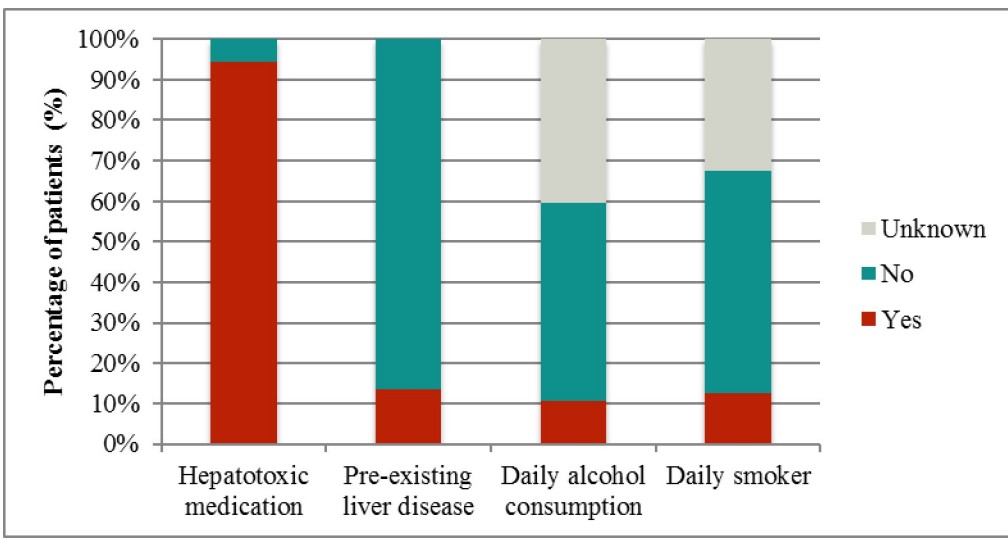

**Fig 1. Risk factors for elevated liver proteins.** Of the total, 94.2% of patients were taking at least one type of medication with hepatotoxic potential; 13.5% had pre-existing liver diseases such as liver cirrhosis, hepatomegaly, hepatitis, cholangitis, steatohepatitis, cholestasis, liver tissue damage, and/or Morbus Hodgkin; 11% are used to daily alcohol consumption and 12% were daily smokers.

4.3 medications with hepatotoxic potential simultaneously). Fourteen patients (13.5%) had pre-existing liver diseases such as liver cirrhosis, hepatomegaly, hepatitis, cholangitis, steatohepatitis, cholestasis, liver tissue damage, and/or Morbus Hodgkin. Smaller percentages of patients for whom data were available engaged in smoking and drinking habits.

**Tumor characteristics.** The average age of patients at time of breast cancer diagnosis was 51.4 years (range, 26–82 years). The average time between initial diagnosis of breast cancer and the detection of BCLM in the cohort was 5.2 years.

Ninety-seven patients (93.3%) had immunohistochemistry data of the metastasis available: 15 (15.7%) had luminal A, 30 (30.9%) luminal B, 32 (33.0%) HER-2 positive, and 20 (20.6%) triple-negative molecular subtypes. The immunohistochemistry results of seven patients were unknown.

Ninety-nine patients (95.2%) had tumor types that were histologically verified. Invasive ductal carcinoma was present in 80.8% of the tumors (n = 80), whereas invasive lobular carcinoma was detected in 17.2% (n = 17). Mixed carcinoma was present in two (2.0%) of the cases. Eighty-four of the patients (80.8%) had metastases in at least one other organ. The most common site of metastasis was bone (n = 65; 77.4% of the 84 with metastases in other organs), followed by lung (n = 40; 47.6%), brain (n = 15; 17.9%), skin (n = 12; 14.3%), peritoneum (n = 4; 4.8%), and ovary (n = 1; 1.2%). The remaining 20 patients (19.2%) had liver metastases only. Thirty-nine patients (37.5% of all patients) had metastases in at least two organs other than the liver. The median largest intrahepatic lesion was 20 mm (mean = 28.9 mm). Solitary lesions were found in 14 patients (14.1%): 34 (34.3%) had 2–5 metastases, 11 (11.1%) had 6–10 metastases, and 40 (40.4%) exhibited a miliary pattern.

**Liver function indicator values.** Liver function test results are summarized in Fig 2.

*AST.* Six months before the diagnosis of BCLM, 19 out of 68 patients (27.9%) had elevated AST levels, which was significantly more than the 5% initially expected ($p < 0.001$). Among them, the median AST level was 26.0 U/L (mean 34.2 U/L). At the time of diagnosis with BCLM, AST levels were significantly higher than 6 months before with a median of 36.0 U/L (mean 65.4 U/L; $p < 0.001$). Twelve months after diagnosis, AST levels remained higher when

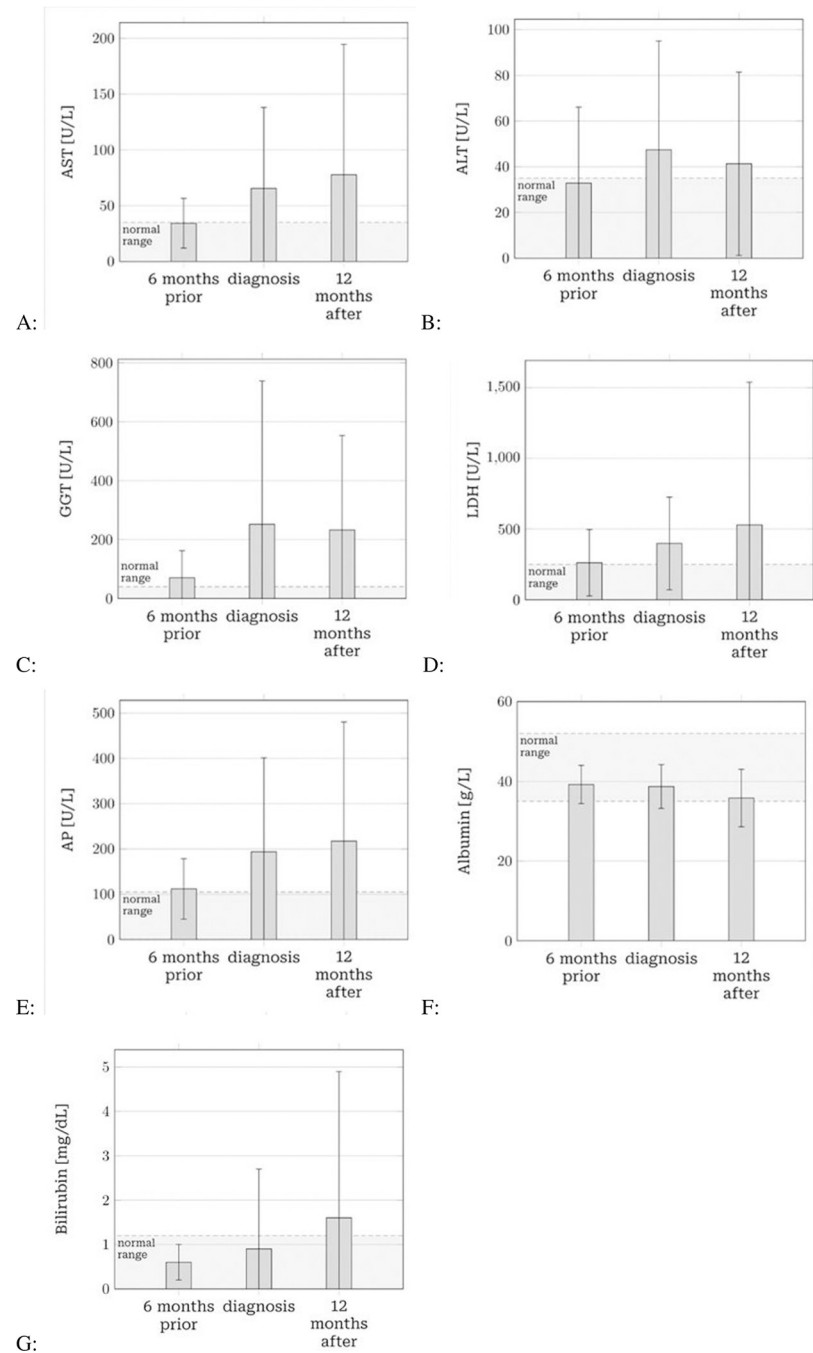

**Fig 2. Mean liver function protein level changes over time.** Mean liver function protein level changes over time. The concentrations (Y-axis) of each enzyme, including AST (A), ALT (B), GGT (C), LDH (D), AP (E), and proteins, including Albumin (F), and Bilirubin (G), are plotted as bar graph, showing values 6-month prior, at the time of diagnosis, and 12-month post-diagnosis, represented on X-axis. The light grey belt shows normal ranges of protein level. Statistical analysis was performed using SPSS software and the p-values are represented as significant with a level of 0.05. After corrected by the Bonferroni method given the number of 21 tests, yielding a significance level of α = 0.0024.

compared to six months prior to diagnosis, with a median of 35 U/L (mean 77.7 U/L). This difference was not significant after correction ($p = 0.004$).

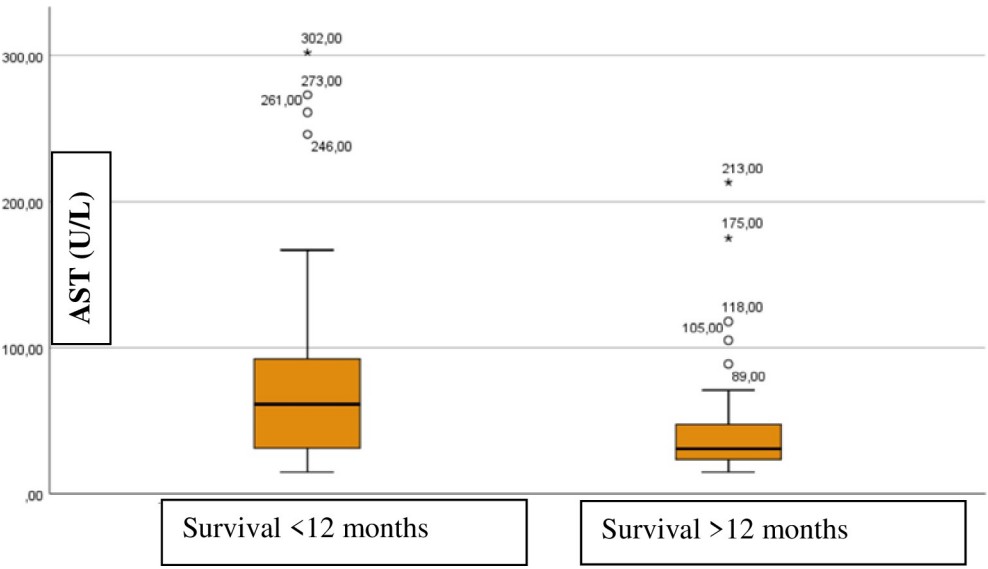

**Fig 3. Correlation between aspartate aminotransferase levels at the time of diagnosis with breast cancer liver metastasis and its dependency on survival.** Patients who died during the period of observation (12 months after diagnosis of breast cancer associated liver metastasis) had significantly higher aspartate aminotransferase levels at the time of diagnosis, than those who survived the study period ($p = 0.002$).

The number of liver metastases (encoded by the four ordinal levels reported above) was positively correlated with AST levels at the time of BCLM diagnosis (Spearman r = 0.525, $p<0.001$). Moreover, we found that mortality was dependent on the AST level, as patients who died during the period of observation (12 months after diagnosis of BCLM) had significantly higher AST levels at the time of diagnosis than those who survived the study period ($p = 0.002$) (Fig 3).

No significant correlations or dependencies were found between increased AST levels and different molecular subtypes, tumor types, grading, alcohol consumption, smoking, BMI, age at initial diagnosis of breast cancer, or age at the time of BCLM diagnosis.

*ALT*. Six months before the diagnosis of BCLM, the median ALT level was 22 U/L (mean 32.8 U/L). Nineteen out of 69 patients (27.5%) had levels above normal range, again significantly more than the expected 5% ($p<0.001$). At the time of diagnosis the median ALT level was 30 U/L (mean 47.4 U/L), which was found to not be significantly different from the timepoint before diagnosis of BCLM after correction ($p = 0.006$). Twelve months after diagnosis, ALT was found to exhibit a median of 28 U/L (mean 41.3 U/L), which was above levels from 6 months prior to diagnosis ($p = 0.083$).

Similar to AST, a positive correlation between the number of metastatic lesions and ALT level at the time of diagnosis was found (Spearman r = 0.418, $p<0.001$).

*GGT*. Six months before diagnosis of BCLM, GGT levels were elevated in 29 out of 69 patients (42.0%), with a median of 36.0 U/L (mean, 70.3 U/L). This was significantly higher than the expected 5% in a healthy population ($p < 0.001$). At the time of BCLM diagnosis, GGT levels showed a median of 53 U/L (mean 251.9 U/L) and were above the normal range in 63.1% of all patients. After 12 months, the mean GGT was found to be lower with a median of 83 U/L (mean 232.7 U/L). At both time points levels were significantly higher than 6 months prior to diagnosis ($p<0.001$).

GGT levels at the time of diagnosis were significantly correlated with the number of metastatic lesions (Spearman r = 0.494, $p<0.001$).

*LDH*. With a median of 198.0 U/L (mean 261.2 U/L) 6 months prior to BCLM, LDH levels were above normal range in 16 out of 58 patients (27.6%), which is significantly more than the expected 5%. At the time of diagnosis of BCLM, a median LDH level of 266 U/L (mean 398.0 U/L) was found. A median value of 237.5 (mean 529.5 U/L) for the timepoint after 12 months was determined. Levels at the time of diagnosis were significantly higher than 6 months prior to diagnosis ($p<0.001$), but not 12 months after diagnosis ($p = 0.17$).

LDH levels at the time of diagnosis were significantly correlated with the number of metastatic lesions (Spearman r = 0.268, $p = 0.012$).

*Albumin*. Six months prior to BCLM diagnosis, albumin levels were below normal range in 8 out of 48 cases (16,7%), which was significantly higher than the expected 2.5%, with a median level of 39.6 U/L (mean 39.2 U/L). The median and mean levels at the time of diagnosis and 12 months thereafter remained similar but decreased (39.6 and 38.7, as well as 36.45 and 35.8, respectively), whereas the percentage of patients below the normal range increased to 24.7% and 40.3%, respectively. The decrease after 12 months was not significant after correction ($p = 0.015$) when compared to 6 months prior to diagnosis.

Patients who survived less than 12 months after being diagnosed with BCLM had significantly lower levels of albumin at the time of diagnosis than patients with an overall survival of >12 months ($p = 0.002$).

*AP*. AP levels were above normal range in 27 out of 68 patients (39.7%) 6 months prior to BCLM diagnosis, which was significantly more than the expected 5% in a normal population ($p<0.001$). It showed a median of 93 U/L (mean 111.9 U/L). The levels exhibited had a median of 113 U/L (mean 193,7 U/L) and 116 U/L (mean 217.4 U/L) at the time of diagnosis and 12 months thereafter, respectively. Differences between timepoints were not significant after correction ($p = 0.17$ and $p = 0.008$, respectively).

AP levels at the time of diagnosis were significantly correlated with the number of metastatic lesions (Spearman r = 0.405, $p<0.001$).

*Bilirubin*. Five out of 68 patients (7.4%) had bilirubin levels above the normal range, which was found to not be significantly different from the expected 5% threshold ($p = 0.373$). The percentage of patients exhibiting elevated bilirubin levels was shown to be significantly higher (12.9%; 22 out of 89, $p<0.001$) at the time of diagnosis, and 12 months after diagnosis (19.2%; 14 out of 73, $p<0.001$), while medians (0.47 U/L 6 months prior, 0.47 U/L at time of diagnosis, 0.49 U/L 12 months thereafter) and means (0.6, 0.9, 1.6 U/L, respectively) did not increase significantly ($p = 0.582$ and $p = 0.012$, respectively).

Bilirubin levels at the time of diagnosis were significantly correlated with the number of metastatic lesions (Spearman r = 0.255, $p = 0.012$).

## Discussion

Our analysis revealed that 40.2% of patients exhibited a significant elevation in GGT levels 6 months prior to diagnosis. The level increased at the time of BCLM and correlated with the number of metastases present. BCLM and increased GGT and AP shows increased biliary injury.

GGT is the most sensitive liver enzyme, even though its fluctuations lack specificity [16]. Additionally, nonalcoholic fatty liver disease (NAFLD) and cholestasis, both of which are possible causes of increased GGT values, share some of the same risk factors as breast cancer [17]. Interestingly, a case-control study of 540 patients by Kwak et al. found that NAFLD was significantly associated with breast cancer in non-obese women; this might be one explanation for the elevated GGT levels observed in our study and is therefore a confounder.

Even though it is limited to a single organ, liver metastasis indicates poor prognosis [18].

Cao et al. reported the median survival time with liver metastasis to be 11.2 months [14]. Furthermore, O'Reilly et al. showed a correlation between elevated AST levels and mortality [13], while Cheung et al. described the correlation of albumin levels with shorter survival time [19]. In addition to elevated AST, low albumin levels were significantly associated with earlier death in our study. Patients who survived for less than 12 months after being diagnosed with BCLM had significantly lower levels of albumin ($p = 0.002$).

The implication of the normal bilirubin at baseline, i.e. 6 months before BCLM diagnosis, despite all other LFTs reported here being elevated should be discussed. This observation is potentially due to hepatocellular damage rather than cholestasis.

Liver enzyme levels can be a good indicator of BCLM. Therefore, to improve medical treatment, with new therapy options like selective internal radiation therapy [20], and long-term prognosis, we recommend screening for liver metastases in women with breast cancer who have elevated liver enzyme levels.

The increasing elevation of AST, LDH, AP, and bilirubin, as well as the decreasing albumin levels 12 months after BCLM diagnosis could be because of disease recurrence or the hepatotoxic drugs given.

## Limitations

Our study was limited owing to its small sample size. Hepatotoxic medications and pre-existing liver disease, such as hepatitis B infection [21], and NAFLD in patients is a potential confounder. The same is true for the patients with bone lesions which could elevate AP. We are planning a large-cohort prospective study in breast cancer patients with liver metastases, including AP isoenzyme levels to see how much AP is from bone tissue and how much from the liver. Nevertheless, this is the first study to investigate liver protein level changes before versus after detection of BCLM.

## Conclusion

We recommend performing liver function tests in women with breast cancer as a screen for potential BCLM. Due to the new treatment options available, the survival time could increase. A prospective multicenter study should be conducted to verify our findings and to determine suitable intervals of checking liver function-associated protein levels during follow-up care.

## Supporting information

**S1 File. Additional information about mean, standard deviation, median and patients included for each time point and liver function protein.**
(DOCX)

## Author Contributions

**Conceptualization:** Carmen Leser, Maximilian Marhold, Christian Singer, Daphne Gschwantler-Kaulich.

**Data curation:** Carmen Leser, Anemone Rutter, Mert Döger, Christine Deutschmann, Iris Holzer.

**Formal analysis:** Carmen Leser, Georg Dorffner.

**Methodology:** Iris Holzer.

**Resources:** Carmen Leser, Maximilian Marhold.

**Supervision:** Daphne Gschwantler-Kaulich.

**Writing – original draft:** Carmen Leser.

**Writing – review & editing:** Deirdre Maria König-Castillo, Christine Deutschmann, Iris Holzer, Daniel König-Castillo.

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
