## [Decision Letter · Decision Letter 0]

20 Jul 2022

PONE-D-22-13541Liver function indicators in patients with breast cancer before and after detection of hepatic metastasesPLOS ONE

Dear Dr. Leser,

Thank you for submitting your manuscript to PLOS ONE. After careful consideration, we feel that it has merit but does not fully meet PLOS ONE’s publication criteria as it currently stands. Therefore, we invite you to submit a revised version of the manuscript that addresses the points raised below by the reviewers during the review process. Please submit your revised manuscript by Sep 03 2022 11:59PM. If you will need more time than this to complete your revisions, please reply to this message or contact the journal office at plosone@plos.org. Please include the following items when submitting your revised manuscript:A rebuttal letter that responds to each point raised by the academic editor and reviewer(s). You should upload this letter as a separate file labeled 'Response to Reviewers'.A marked-up copy of your manuscript that highlights changes made to the original version. You should upload this as a separate file labeled 'Revised Manuscript with Track Changes'.An unmarked version of your revised paper without tracked changes. You should upload this as a separate file labeled 'Manuscript'.

We look forward to receiving your revised manuscript.

Kind regards,

Surinder K. Batra

Academic Editor

PLOS ONE

Journal Requirements:

Reviewers' comments:

Reviewer's Responses to Questions

**Comments to the Author**

1. Is the manuscript technically sound, and do the data support the conclusions?

Reviewer #1: Partly

Reviewer #2: Partly

2. Has the statistical analysis been performed appropriately and rigorously? 

Reviewer #1: Yes

Reviewer #2: Yes

3. Have the authors made all data underlying the findings in their manuscript fully available?

Reviewer #1: Yes

Reviewer #2: Yes

4. Is the manuscript presented in an intelligible fashion and written in standard English?

Reviewer #1: Yes

Reviewer #2: No

5. Review Comments to the Author

Reviewer #1: In Liver function indicators in patients with breast cancer before and after detection of hepatic metastases, the authors set out to retrospectively assess LFTs, in particular proteins, as well as demographic and tumour characteristics over 18 months in breast cancer patients w/metastatic liver disease (BCLM). They documented values at 3 time points: 6 months before BCLM diagnosis, at BCLM diagnosis, and 12 months after BCLM diagnosis in 104 patients gathered over 39 years.

They found very significant (p<.001) elevations in AST, ALT, GGT, LDH and AP, and similarly significant (p<.001) decrease in albumin. These LFT abnormalities were independent of patient- or tumour-specific factors. Furthermore, AST and albumin levels were significantly (p <.002) associated with decreased OS

General Comments Given the prevalence of breast cancer and the poor clinical course once liver metastases have developed, this manuscript covers a highly relevant topic.

Specific Comments

1. Only 69 + 40= 109 patients had labs before and at time of dx. Yet your cohort is 104 rather than 109. Why were 5 of these patients excluded?

2. Hepatotoxic meds in the 98 patients is a potential confounder, i.e. if LFTs elevated, it could be either due to BCLM or DILI. The same is true for the 14 patients w/pre-existing liver disease.

3. In those w/BCLM and bone metastases, the bone lesions are also a confounder because they could elevate AP. Was the AP isoenzyme test done to see how much of the AP came from bone and how much from liver? This would be helpful if a prospective study is done to correlate elevated LFTs with BCLM.

4. Although you found a correlation between the number of liver metastases and the degree of elevation of the labs, unless the vast majority of your already small cohort were on the same liver-toxic drugs, the differences in lab values could be due to differences in drug toxicities rather than more advanced BCLM, i.e. greater number of liver metastases. A larger cohort is probably needed to untangle this data.

5. On page 7, under GGT, you refer to your cohort as a “healthy population.” That contradicts the data you reported above (bottom p 5 /top p 6) regarding the number of patients on hepatotoxic drugs (ca 94%), and the number of patients w/metastases involving at least one other organ (ca 80%). That could mean the cohort already had DILI and/or micrometastases to the liver in the months preceding BCLM diagnosis.

6. Again, on p 8, it should not be so surprising that the AP was elevated before BCLM diagnosis since you mentioned above that a majority of the cohort had bone metastases.

7. Please modify Table 2, subdividing each of the 3 columns into 4, then put the appropriate heading above the new sub columns, i.e. number of BCLM patients; mean lab value; what is the 3rd sub column, i.e. (22.3) under AST?; median lab value

8. From Table 2, how do you explain the increasing elevation of AST, LDH, AP, and bilirubin, as well as the decreasing albumin 12 months after BCLM diagnosis? Is this because the patients were not treated once BCLM diagnosed or were they treated and the rise represents disease recurrence or post-treatment effect from hepatotoxic drugs? At least speculate on the cause.

9. Please explain, in the Discussion, the implication of the normal bilirubin at baseline, i.e. 6 months before BCLM diagnosis, although all of the other LFTs reported here are elevated. In other words, what does this mean about the type of liver damage that the drugs may be causing? I think the damage is probably hepatocellular rather than cholestatic.

10. In the Discussion, please clarify why NAFLD was associated w/breast cancer? Is this due to underlying NAFLD or the effect of chemotherapy? This is important as NAFLD is then another confounder in LFT evaluation in these patients.

11. In the Abstract, “may improve both their early prediction and treatment,” should be “may improve both their early detection and treatment.” OR “may improve both their prediction and treatment.”

12. In Intro, first paragraph, “the incidence rate of metastasis” should be simply “the incidence of metastases”.

Reviewer #2: In the manuscript titled “Liver function indicators in patients with breast cancer before and after detection of hepatic metastases”, Leser et al. have reported a clinically relevant prognostic significance of liver functions in breast cancer liver metastasis. A retrospective temporal analysis of different liver enzymes Aspartate aminotransferase, alanine aminotransferase, gamma-glutamyltransferase, lactate dehydrogenase and alkaline phosphatase, and albumin protein are important parameters to predict liver metastasis in breast cancer patients. However, it is expected that the authors generate more figures from the data compiled in the tabular form for a better understanding of the results. In addition, the correlation plot must be shown if both aminotransferase and albumin correlate with the patient’s survival. Irrespective of significance, both univariate and multivariate analysis must be presented to find the significance and correlation of proposed prognostic liver function proteins/enzymes. Unfortunately, the current form of the manuscript is not acceptable. However, the following are comments that must be addressed so that the manuscript can be reconsidered for publication in PLOS One. Please see the following comments:

1. Add continuous line numbers in the manuscript so that comments can be addressed to the page and line number.

2. References cited do not cover the recent publications in the field. For example, references for statistical values presented in the introduction are old, including refs.1-4. Please include the most up-to-date information with recent citations.

3. It is not clear what the treatment was for the patients diagnosed with liver metastasis for 12 months post-diagnosis. Is there information available on whether these patients were considered for surgery or any other treatment modality?

4. Description of each enzyme and protein has been given in the results. However, authors are suggested to plot the values to make them clearer and more understandable.

5. Baseline value for each parameter in normal individuals can be merged in table 2, and data can be plotted in the form of bar graphs.

6. Figure legends are missing for figures given in the manuscript. Please provide the figure legends, including the statistical significance.

7. Font size and type are not consistent in eh manuscript and figures. Authors are suggested to revise the manuscript with the same font type and size as per the journal’s guidelines.

8. It has not been mentioned what the label on the Y-axis of figure 2 is. Please add the Y-axis details.

9. Discussion needs to be elaborated with overall more citations of recent work in the field.

10. Authors are suggested to explain why enzyme and protein levels went down after 12 months of diagnosis of BCLM, trending towards the values observed 6 months prior to the diagnosis of BCLM. Is this an effect of treatment in these patients? Were there patients where progressive liver metastases showed a positive correlation with elevated enzyme and protein levels?

11. Page 8, Line 8, the word prediction should be replaced by diagnosis.

12. Page 8, Is the statement “Departments of Internal Medicine I and Gynecology at..” correctly mentioned in the method section of the abstract.

13. Page 8: The sentence needs to be ended after the "…detection of liver metastases (p<0.001)”. Please add a dot in the result section of the abstract or connect the following sentence properly.

6. PLOS authors have the option to publish the peer review history of their article (what does this mean?). If published, this will include your full peer review and any attached files.

Reviewer #1: No

Reviewer #2: **Yes: **Shailendra K Gautam

---

## [Author Response · Author response to Decision Letter 0]

26 Aug 2022

Dear Editor: 

We/I wish to re-submit the manuscript titled “Liver function indicators in patients with breast cancer before and after detection of hepatic metastases.” The manuscript ID is PONE-D-22-13541.

We thank you and the reviewers for your thoughtful suggestions and insights. The manuscript has benefited from these insightful suggestions. I look forward to working with you and the reviewers to move this manuscript closer to publication in the PLoS ONE.

The manuscript has been rechecked and the necessary changes have been made in accordance with the reviewers’ suggestions. The responses to all comments have been prepared and attached herewith. 

Thank you for your consideration. I look forward to hearing from you.

Sincerely,

Leser Carmen, MD, PhD

Address: Waehringer Guertel 18-20, 1090 Vienna, Austria

Phone: 0043/660/2040087

Fax: 0043/1/40400 23230

Email: carmen.leser@meduniwien.ac.at

Reviewer #1: In Liver function indicators in patients with breast cancer before and after detection of hepatic metastases, the authors set out to retrospectively assess LFTs, in particular proteins, as well as demographic and tumour characteristics over 18 months in breast cancer patients w/metastatic liver disease (BCLM). They documented values at 3 time points: 6 months before BCLM diagnosis, at BCLM diagnosis, and 12 months after BCLM diagnosis in 104 patients gathered over 39 years.

They found very significant (p<.001) elevations in AST, ALT, GGT, LDH and AP, and similarly significant (p<.001) decrease in albumin. These LFT abnormalities were independent of patient- or tumour-specific factors. Furthermore, AST and albumin levels were significantly (p <.002) associated with decreased OS

General Comments Given the prevalence of breast cancer and the poor clinical course once liver metastases have developed, this manuscript covers a highly relevant topic.

Specific Comments

1. Only 69 + 40= 109 patients had labs before and at time of dx. Yet your cohort is 104 rather than 109. Why were 5 of these patients excluded? 

Response: Five patients were excluded due to a lack of data.

2. Hepatotoxic meds in the 98 patients is a potential confounder, i.e. if LFTs elevated, it could be either due to BCLM or DILI. The same is true for the 14 patients w/pre-existing liver disease. Response: This has been mentioned in the limitations.

3. In those w/BCLM and bone metastases, the bone lesions are also a confounder because they could elevate AP. Was the AP isoenzyme test done to see how much of the AP came from bone and how much from liver? This would be helpful if a prospective study is done to correlate elevated LFTs with BCLM. 

Response: This is planned, and we have mentioned bone lesions as a confounder.

4. Although you found a correlation between the number of liver metastases and the degree of elevation of the labs, unless the vast majority of your already small cohort were on the same liver-toxic drugs, the differences in lab values could be due to differences in drug toxicities rather than more advanced BCLM, i.e. greater number of liver metastases. A larger cohort is probably needed to untangle this data. 

Response: A prospective study has been planned.

5. On page 7, under GGT, you refer to your cohort as a “healthy population.” That contradicts the data you reported above (bottom p 5 /top p 6) regarding the number of patients on hepatotoxic drugs (ca 94%), and the number of patients w/metastases involving at least one other organ (ca 80%). That could mean the cohort already had DILI and/or micrometastases to the liver in the months preceding BCLM diagnosis. 

Response: Thank you for your comment. I meant that it was higher than in a healthy population, and of course this could be because of your annotations. I have rewritten it to improve clarity. 

6. Again, on p 8, it should not be so surprising that the AP was elevated before BCLM diagnosis since you mentioned above that a majority of the cohort had bone metastases. 

Response: I have rewritten it to improve clarity.

7. Please modify Table 2, subdividing each of the 3 columns into 4, then put the appropriate heading above the new sub columns, i.e. number of BCLM patients; mean lab value; what is the 3rd sub column, i.e. (22.3) under AST?; median lab value 

Response: I have included these details in a figure.

8. From Table 2, how do you explain the increasing elevation of AST, LDH, AP, and bilirubin, as well as the decreasing albumin 12 months after BCLM diagnosis? Is this because the patients were not treated once BCLM diagnosed or were they treated and the rise represents disease recurrence or post-treatment effect from hepatotoxic drugs? At least speculate on the cause. 

Response: Thank you for your comment. I have included this in the manuscript.

9. Please explain, in the Discussion, the implication of the normal bilirubin at baseline, i.e. 6 months before BCLM diagnosis, although all of the other LFTs reported here are elevated. In other words, what does this mean about the type of liver damage that the drugs may be causing? I think the damage is probably hepatocellular rather than cholestatic. 

Response: Thank you for your comment. I have included this in the manuscript.

10. In the Discussion, please clarify why NAFLD was associated w/breast cancer? Is this due to underlying NAFLD or the effect of chemotherapy? This is important as NAFLD is then another confounder in LFT evaluation in these patients. 

Response: We have yet to clarify the cause, and so we have included it as a confounder.

11. In the Abstract, “may improve both their early prediction and treatment,” should be “may improve both their early detection and treatment.” OR “may improve both their prediction and treatment.” – 

Response: Thank you for your comment. I have changed this accordingly.

12. In Intro, first paragraph, “the incidence rate of metastasis” should be simply “the incidence of metastases”. – 

Response: Thank you for your comment. I have changed this accordingly.

 

Reviewer #2: In the manuscript titled “Liver function indicators in patients with breast cancer before and after detection of hepatic metastases”, Leser et al. have reported a clinically relevant prognostic significance of liver functions in breast cancer liver metastasis. A retrospective temporal analysis of different liver enzymes Aspartate aminotransferase, alanine aminotransferase, gamma-glutamyltransferase, lactate dehydrogenase and alkaline phosphatase, and albumin protein are important parameters to predict liver metastasis in breast cancer patients. However, it is expected that the authors generate more figures from the data compiled in the tabular form for a better understanding of the results. In addition, the correlation plot must be shown if both aminotransferase and albumin correlate with the patient’s survival. Irrespective of significance, both univariate and multivariate analysis must be presented to find the significance and correlation of proposed prognostic liver function proteins/enzymes. Unfortunately, the current form of the manuscript is not acceptable. However, the following are comments that must be addressed so that the manuscript can be reconsidered for publication in PLOS One. Please see the following comments:

1. Add continuous line numbers in the manuscript so that comments can be addressed to the page and line number. 

Response: Line numbers have been added to the manuscript.

2. References cited do not cover the recent publications in the field. For example, references for statistical values presented in the introduction are old, including refs.1-4. Please include the most up-to-date information with recent citations. 

Response: Thank you for your comment. I have updated the references accordingly.

3. It is not clear what the treatment was for the patients diagnosed with liver metastasis for 12 months post-diagnosis. Is there information available on whether these patients were considered for surgery or any other treatment modality? 

Response: None of them received surgery. They all remained on the medication-based therapy, but the medication which was given varied. 

4. Description of each enzyme and protein has been given in the results. However, authors are suggested to plot the values to make them clearer and more understandable. 

Response: We request to not make this change because we believe that these revisions will not help to better understand the data. However, we have included the results in Table 2.

5. Baseline value for each parameter in normal individuals can be merged in table 2, and data can be plotted in the form of bar graphs. 

Response: We have added a figure to make the results more understandable. 

6. Figure legends are missing for figures given in the manuscript. Please provide the figure legends, including the statistical significance. 

Response: Thank you for your comment. We have added the figure legends and included the statistical significance.

7. Font size and type are not consistent in eh manuscript and figures. Authors are suggested to revise the manuscript with the same font type and size as per the journal’s guidelines. 

Response: The font has been changed to Times New Roman throughout the manuscript.

8. It has not been mentioned what the label on the Y-axis of figure 2 is. Please add the Y-axis details. Response: The Y-axis details have been included.

9. Discussion needs to be elaborated with overall more citations of recent work in the field. Response: I have tried to include more recent work, but there have been no recent publications in that field worth mentioning.

10. Authors are suggested to explain why enzyme and protein levels went down after 12 months of diagnosis of BCLM, trending towards the values observed 6 months prior to the diagnosis of BCLM. Is this an effect of treatment in these patients? Were there patients where progressive liver metastases showed a positive correlation with elevated enzyme and protein levels? 

Response: These details have been included.

11. Page 8, Line 8, the word prediction should be replaced by diagnosis. 

Response: Thank you for your comment. I have made this change.

12. Page 8, Is the statement “Departments of Internal Medicine I and Gynecology at..” correctly mentioned in the method section of the abstract. 

Response: Thank you for your comment. I have made this change.

13. Page 8: The sentence needs to be ended after the "…detection of liver metastases (p<0.001)”. Please add a dot in the result section of the abstract or connect the following sentence properly. Response: I have included this accordingly.

---

## [Decision Letter · Decision Letter 1]

25 Oct 2022

PONE-D-22-13541R1Liver function indicators in patients with breast cancer before and after detection of hepatic metastasesPLOS ONE

Dear Dr. Leser,

Thank you for submitting your manuscript to PLOS ONE. After careful consideration, we feel that it has merit but does not fully meet PLOS ONE’s publication criteria as it currently stands. Therefore, we invite you to submit a revised version of the manuscript that addresses the minor points raised by the reviewers during the review process. Please submit your revised manuscript by Dec 09 2022 11:59PM. If you will need more time than this to complete your revisions, please reply to this message or contact the journal office at plosone@plos.org. Please include the following items when submitting your revised manuscript:A rebuttal letter that responds to each point raised by the academic editor and reviewer(s). You should upload this letter as a separate file labeled 'Response to Reviewers'.A marked-up copy of your manuscript that highlights changes made to the original version. You should upload this as a separate file labeled 'Revised Manuscript with Track Changes'.An unmarked version of your revised paper without tracked changes. You should upload this as a separate file labeled 'Manuscript'.If applicable, we recommend that you deposit your laboratory protocols in protocols.io to enhance the reproducibility of your results. Protocols.io assigns your protocol its own identifier (DOI) so that it can be cited independently in the future. For instructions see: https://journals.plos.org/plosone/s/submission-guidelines#loc-laboratory-protocols. Additionally, PLOS ONE offers an option for publishing peer-reviewed Lab Protocol articles, which describe protocols hosted on protocols.io. Read more information on sharing protocols at https://plos.org/protocols?utm_medium=editorial-email&utm_source=authorletters&utm_campaign=protocols.

We look forward to receiving your revised manuscript.

Kind regards,

Surinder K. Batra

Academic Editor

PLOS ONE

Journal Requirements:

Reviewers' comments:

Reviewer's Responses to Questions

**Comments to the Author**

1. If the authors have adequately addressed your comments raised in a previous round of review and you feel that this manuscript is now acceptable for publication, you may indicate that here to bypass the “Comments to the Author” section, enter your conflict of interest statement in the “Confidential to Editor” section, and submit your "Accept" recommendation.

Reviewer #2: All comments have been addressed

Reviewer #3: (No Response)

2. Is the manuscript technically sound, and do the data support the conclusions?

Reviewer #2: Yes

Reviewer #3: Yes

3. Has the statistical analysis been performed appropriately and rigorously? 

Reviewer #2: Yes

Reviewer #3: Yes

4. Have the authors made all data underlying the findings in their manuscript fully available?

Reviewer #2: Yes

Reviewer #3: Yes

5. Is the manuscript presented in an intelligible fashion and written in standard English?

Reviewer #2: Yes

Reviewer #3: Yes

6. Review Comments to the Author

Reviewer #2: Overall, the manuscript looks good. Authors have revised the manuscript and addressed the comments and concerns satisfactorily. The revised manuscript can be considered for publication in the journal PLOS One. However, few more corrections need to be made to further improve the manuscript:

1. Each enzyme or protein plot can be relabeled in alfa-numeric format in the Figure 2 and accordingly it can be mentioned in the figure legend and result section. Authors are suggested to label each graph from 2A-2G.

2. Legend and result have been combined for figure 2, which must be separated (Line 162-179). The description given in the revised manuscript is fitting well as result. Authors are advised to write figure legend after labeling each graph, as suggested in comment 1. Moreover, Authors must consider revising the legends of each figure, as the purpose of legend is to describe the figure, its axes, dimensions, and statistics, not to explain the results. Please consider revising the legends following legend template given below for figure 2:

Figure 2: Mean liver function protein level changes over time. The concentrations (Y-axis) of each enzyme, including AST (A), ALT (B), GGT (C), LDH (D), AP (E), and proteins, including Albumin (F), and Bilirubin (G), are plotted as bar graph, showing values 6-month prior, at the time of diagnosis, and 12-month post-diagnosis, represented on X-axis. Statistical analysis was performed using………..software and the p-values are represented as…………….

Reviewer #3: This is a revision manuscript entitled “Liver function indicators in patients with breast cancer before and after detection of hepatic metastases” led by Carmen et al., and have significantly revised based on previous concerns raised by the reviewers. The manuscript is highly relevant topic given the poor clinical course of liver metastasis with breast cancer patients.

Minor comments to improve the manuscript.

a. Authors can consider revising their title to reflect that this is a retrospective

b. Also, BCLM and increased GGT and AP show that there increased biliary injury can be added in the manuscript. Further biliary injury has correlated with metastatic lesions has been observed in the manuscript

c. Table 2 is confusing; number of samples can be in a separate column. Also, a dotted plot would with individual data would be a great addition to enhance the clarity of the data presented

d. Authors can add the details of obesity or NAFLD as a confounding variable in their analysis, if they have the data otherwise mentioning them in the limitations is fine

e. Consider adding a subtitle of limitations before conclusion.

f. Figure 2 legends needs the details of sample number and statistical analysis

7. PLOS authors have the option to publish the peer review history of their article (what does this mean?). If published, this will include your full peer review and any attached files.

Reviewer #2: **Yes: **Shailendra K Gautam

Reviewer #3: No

---

## [Author Response · Author response to Decision Letter 1]

4 Nov 2022

We thank you and the reviewers for your thoughtful suggestions and insights. The manuscript has benefited from these insightful suggestions. I look forward to working with you and the reviewers to move this manuscript closer to publication in the PLoS ONE.

The manuscript has been rechecked and the necessary changes have been made in accordance with the reviewers’ suggestions. The responses to all comments have been prepared and attached herewith.

---

## [Decision Letter · Decision Letter 2]

17 Nov 2022

Liver function indicators in patients with breast cancer before and after detection of hepatic metastases - a retrospective study

PONE-D-22-13541R2

Dear Dr. Leser,

We’re pleased to inform you that your manuscript has been judged scientifically suitable for publication and will be formally accepted for publication once it meets all outstanding technical requirements.

Kind regards,

Surinder K. Batra

Academic Editor

PLOS ONE

Additional Editor Comments (optional):

Reviewers' comments:

Reviewer's Responses to Questions

**Comments to the Author**

1. If the authors have adequately addressed your comments raised in a previous round of review and you feel that this manuscript is now acceptable for publication, you may indicate that here to bypass the “Comments to the Author” section, enter your conflict of interest statement in the “Confidential to Editor” section, and submit your "Accept" recommendation.

Reviewer #2: All comments have been addressed

Reviewer #3: All comments have been addressed

2. Is the manuscript technically sound, and do the data support the conclusions?

Reviewer #2: Yes

Reviewer #3: Yes

3. Has the statistical analysis been performed appropriately and rigorously? 

Reviewer #2: Yes

Reviewer #3: Yes

4. Have the authors made all data underlying the findings in their manuscript fully available?

Reviewer #2: Yes

Reviewer #3: Yes

5. Is the manuscript presented in an intelligible fashion and written in standard English?

Reviewer #2: Yes

Reviewer #3: Yes

6. Review Comments to the Author

Reviewer #2: All the comments have been addressed satisfactorily and the manuscript can be considered for publication.

Reviewer #3: Authors have satisfactorily improved their manuscript based on all three reviewers comments.

Manuscript has improved significantly

7. PLOS authors have the option to publish the peer review history of their article (what does this mean?). If published, this will include your full peer review and any attached files.

Reviewer #2: **Yes: **Shailendra K Gautam

Reviewer #3: No

---

## [Editor Report · Acceptance letter]

15 Dec 2022

PONE-D-22-13541R2 

Liver function indicators in patients with breast cancer before and after detection of hepatic metastases-a retrospective study 

Dear Dr. Leser:

I'm pleased to inform you that your manuscript has been deemed suitable for publication in PLOS ONE. Congratulations! Your manuscript is now with our production department. 

Kind regards, 

on behalf of

Prof. Surinder K. Batra 

Academic Editor

PLOS ONE